# Variation of metabolic and bariatric surgery utilization by neighborhood socioeconomic status in Maryland

Oluwasegun Akinyemi[1,2]*, Terrence Fullum[3], Mojisola Fasokun[4], Kakra Hughes[3], Dahai Yue[2], Craig Scott Fryer[2], Jie Chen[2], Kellee White-Whilby[2]

1 The Clive O. Callender Outcomes Research Center, Howard University College of Medicine, Washington District of Columbia, United States of America, 2 Department of Health Policy and Management, University of Maryland School of Public Health, College Park, United States of America, 3 Department of Surgery, Howard University College of Medicine, Washington District of Columbia, United States of America, 4 Department of Epidemiology, University of Alabama at Birmingham, Birmingham, Alabama, United States of America

* austineakinyemi@gmail.com

## Abstract

### Importance

Metabolic and Bariatric Surgery (MBS) is a proven treatment for severe obesity, yet disparities in its utilization persist, particularly among socioeconomically disadvantaged populations.

### Objective

To evaluate the association between neighborhood socioeconomic status (nSES) and MBS utilization in Maryland and assess whether this relationship varies by race and ethnicity.

### Design, setting, and participants

A cross-sectional study using the Maryland State Inpatient Database (2018–2020), linked with the Distressed Communities Index (DCI). The study included adults aged ≥18 years with body mass index (BMI) ≥35 kg/m² who were eligible for MBS. Race/ethnicity was self-reported and categorized as non-Hispanic White, non-Hispanic Black, Hispanic, or Other.

### Main outcomes and measures

The primary outcome was receipt of MBS. The primary exposure was nSES, measured using DCI quintiles (prosperous, comfortable, mid-tier, at-risk, and distressed). Multivariable logistic regression models estimated the adjusted odds of undergoing surgery, accounting for age, sex, race/ethnicity, insurance, comorbidities, obesity class, and urbanicity. Interaction terms tested effect modification by race.

**Data availability statement:** This study utilized data from the Healthcare Cost and Utilization Project (HCUP) State Inpatient Databases (SID), which are maintained by the Agency for Healthcare Research and Quality (AHRQ). These datasets are third-party data that are not publicly available due to legal and licensing restrictions. However, interested research-ers may obtain access to the same datasets directly from HCUP through the HCUP Central Distributor website (https://hcup-us.ahrq.gov/db/nation/nis/nisdde.jsp). Specific information regarding the State Inpatient Databases can be found on the HCUP website (https://hcup-us.ahrq.gov/db/state/siddbdocumentation.jsp). Researchers can replicate the study in its entirety by obtaining the HCUP State Inpatient Databases from the HCUP Central Distributor and applying the methodology described in the paper. The authors did not have any special access privileges to these data beyond those that are available to all qualified researchers who complete the HCUP Data Use Agreement and purchase the datasets.

**Funding:** This project was supported in part by the National Institute on Minority Health and Health Disparities of the National Institutes of Health under Award Number 2U54MD007597. The content is solely the responsibility of the authors and does not necessarily represent the official views of the National Institutes of Health. Dr. Oluwasegun Akinyemi received a salary from Howard University during the study period; however, this salary support was not specifically provided for the conduct of this study. The funders had no role in study design, data collection and analysis, decision to pub-lish, or preparation of the manuscript.

**Competing interests:** The authors have declared that no competing interests exist.

## Results

Of 169,026 eligible individuals, 11,963 (7.1%) received MBS. Most recipients were female (82.6%), with nearly equal representation of Black (46.9%) and White (46.1%) patients. A socioeconomic gradient in utilization was evident: individuals from dis-tressed neighborhoods had 30% lower odds of receiving surgery (OR, 0.70; 95% CI, 0.64–0.76) compared to those in prosperous areas. Odds were similarly reduced for mid-tier (OR, 0.74; 95% CI, 0.70–0.79), at-risk (OR, 0.89; 95% CI, 0.83–0.96), and comfortable (OR, 0.89; 95% CI, 0.84–0.95) neighborhoods. Race moderated this association: across all DCI quintiles, Black individuals were more likely than White individuals to undergo surgery, with marginal effects increasing from 0.90% in pros-perous to 2.10% in distressed areas.

## Conclusions and relevance

MBS remains underutilized among eligible patients, especially those in socioeconom-ically disadvantaged neighborhoods. However, utilization patterns differ by race, with higher odds among Black individuals across all neighborhood strata. These findings highlight the need for targeted interventions to improve equity in obesity treatment access.

## Introduction

Severe obesity represents a significant public health challenge, with far-reaching implications for individual health, healthcare systems, and societal well-being [1,2]. Metabolic and bariatric surgery (MBS), also known as weight loss surgery, has emerged as an effective and evidence-based intervention for managing severe obesity and its associated comorbidities [3,4]. By providing sustained weight loss, improved metabolic profiles, and enhanced quality of life, MBS has proven to be a critical treatment for patients who meet the eligibility criteria of body mass index (BMI) thresholds with coexisting medical conditions [5].

Although the impact of individual income on disparities in healthcare access and health outcomes is well-established, the role of neighborhood socioeconomic status (nSES) is gaining recognition as a significant determinant of health [6,7]. The cumu-lative effects of social and economic disadvantage in distressed neighborhoods can influence health behaviors, access to healthcare, and ultimately, the utilization of critical services such as MBS [8,9]. Measures like the Distressed Communities Index (DCI) provide a comprehensive framework for assessing nSES, categorizing neigh-borhoods based on economic, social, and health metrics [10–12]. However, the asso-ciation between nSES and MBS utilization, particularly when accounting for potential modifiers like race and ethnicity, remains underexplored.

Racial and ethnic disparities in socioeconomic status, healthcare access, and insurance coverage further contribute to inequities in MBS utilization [13,14]. Black and Hispanic populations, who disproportionately reside in socioeconomically

disadvantaged neighborhoods, face structural barriers that limit access to surgical care [15,16]. Although MBS is typically covered by public and private insurance, reimbursement policies may still pose challenges for both patients and providers [17–19]. Patients often face variable out-of-pocket costs depending on their insurance plan, and low-income individuals may delay or forgo surgery due to upfront expenses [20]. In addition, declining reimbursement rates for MBS procedures such as the 32.8% decrease for sleeve gastrectomy between 2010 and 2022 may discourage provider availability or limit surgical capacity in underserved areas [21,22]. Understanding the interplay between nSES and race and ethnicity is critical for addressing these disparities and ensuring equitable access to MBS.

This study aims to evaluate the association between neighborhood socioeconomic status (nSES) and MBS utilization in Maryland, a state characterized by demographically diverse populations and significant socioeconomic variation across communities. As of recent data, approximately 34.3% of adults in Maryland were classified as obese, which is slightly above some reported national averages but consistent with broader trends of rising obesity rates across the U.S [23]. Maryland is located in a region known for its high utilization of MBS, reflecting both a significant disease burden and active engagement in surgical obesity treatment. This regional trend suggests a strong commitment to addressing obesity through surgical interventions [24].

## Methodology

### Study design

This study utilized a cross-sectional design to evaluate the relationship between nSES, as measured by the (DCI, and the likelihood of MBS utilization among eligible patients in Maryland. The analysis also investigated the interaction between race and ethnicity and nSES in influencing MBS utilization. The cross-sectional design is appropriate for assessing population-level associations at a specific time point and is consistent with the study objectives.

### Data source

Data for this study were derived from the Maryland State inpatient datasets [25], a comprehensive statewide administrative dataset that integrates hospital discharge records, demographic information, and socioeconomic indicators. The dataset includes all eligible patients undergoing MBS in Maryland from January 1, 2018, to December 31, 2020. The DCI data [26] was linked to patient-level records based on residential ZIP codes. The DCI, developed by the Economic Innovation Group, is a ZIP code–level composite measure of socioeconomic distress. It integrates seven metrics—including poverty, education, housing vacancy, and employment—to quantify neighborhood disadvantages. DCI scores range from 0 (no distress) to 100 (severe distress).

### Study population

The study population consisted of adults aged 18 years and above who are eligible for MBS based on clinical guidelines, including a body mass index (BMI) ≥40 kg/m² or BMI ≥ 35 kg/m² with comorbid conditions [27]. Patients were identified using International Classification of Diseases, 10th Revision (ICD-10) procedure codes specific to MBS. This study included adults aged 18 years or older who met clinical guidelines for MBS eligibility and were Maryland residents, as identified by ZIP code

### Explanatory variable

The DCI can be operationalized in two forms: as a continuous variable and as a categorical variable. Each community is assigned a DCI score based on its performance across the seven metrics, which serves as a granular measure of socioeconomic distress and enables comparisons across ZIP codes and regions. Alternatively, the DCI is categorized into five distinct tiers based on percentile rankings of ZIP codes. These tiers include Prosperous (0th to 20th percentile),

 

Comfortable (21st to 40th percentile), Mid-Tier (41st to 60th percentile), At-Risk (61st to 80th percentile), and Distressed (81st to 100th percentile). This categorical operationalization facilitates the identification of trends and disparities within socioeconomic strata and simplifies interpretation for policy development and public health research.

### Outcome variable

The outcome, MBS utilization, was defined as undergoing any bariatric procedure during the study period. MBS procedures were identified using ICD-10 procedure codes, which included Roux-en-Y gastric bypass, sleeve gastrectomy, and other approved bariatric interventions. This variable was binary, coded as 1 for patients who underwent surgery and 0 for those who did not.

### Covariates

Covariates included age (continuous and categorized as 18–44, 45–64, ≥65), sex (male/female), and race/ethnicity (White, Black, Hispanic, Other). The "Other" group included Asian/Pacific Islander, Native American, multiracial, or unspecified. Insurance type was classified as self-pay, Medicare, Medicaid, private, or other. Obesity class was based on BMI (Class II: 35–39.9; Class III: ≥40); Class I was excluded. Comorbidity was measured using the Charlson Comorbidity Index (0, 1–2, >2). Urban-rural status included rural, small town, suburban, and urban. Year of surgery was included to adjust for time trends.

### Statistical analysis

Baseline characteristics of the study population were summarized using frequencies and percentages for categorical variables. Differences between patients undergoing bariatric surgery and those who did not were assessed using chi-square tests for categorical variables and t-tests for continuous variables.

Logistic regression was used to examine the association between neighborhood socioeconomic status (nSES), measured by the DCI, and MBS utilization among eligible adults. The fully adjusted model included demographic variables (age, sex, race/ethnicity), clinical characteristics (obesity class, Charlson Comorbidity Index), insurance type, and urban classification to provide geographic context.

To assess whether the association between nSES and MBS utilization varied by race and ethnicity, an interaction term between DCI quintiles and race/ethnicity was incorporated. Marginal effects at representative values were estimated using the margins command in Stata to calculate predicted probabilities of surgery across racial groups and DCI categories. Average marginal effects were reported for improved interpretability of interactions in the nonlinear model.This approach is supported by methodological literature emphasizing the importance of marginal effects for interpreting interactions in nonlinear models, where coefficients cannot be directly interpreted in the same manner as in linear models [28,29].

The final model specification is as follows:
$$\text{Logit}(P(Y=1)) = \alpha + \beta 1(\text{DCI}) + \beta 2(\text{Covariates}) + \beta 3(\text{Race}) + \beta 13(\text{Race x DCI})$$, where Y represents MBS utilization, α is the intercept, and β13 captures the interaction effect of race/ethnicity and DCI.

Sensitivity analyses were performed to evaluate the robustness of findings. These included modeling DCI as a continuous variable, using propensity score matching to address potential selection bias, and substituting the Area Deprivation Index (ADI) as an alternative measure of nSES.

### Ethical considerations

This study used publicly available, de-identified administrative data and was exempt from Institutional Review Board (IRB) oversight by the University of Maryland IRB [Protocol #2284677−1]. Because the analysis involved secondary use of de-identified data, no direct interaction with human subjects occurred, and informed consent was not required. Data linkage was performed using encrypted identifiers to protect patient confidentiality and ensure compliance with ethical standards for human subjects research.

## Results

### Baseline characteristics

Of the 169,026 individuals eligible for MBS in Maryland, only 7.1% (n = 11,963) underwent the procedure, while 92.9% (n = 157,063) did not (Table 1).

Patients who received MBS were significantly younger, with a mean age of 44.4 ± 11.8 years compared to 56.7 ± 15.3 years for non-utilizers (p < 0.001). Over half (51.2%) of the bariatric cohort were aged 18–44 years, compared to just 24.2% of non-utilizers. Only 5.1% of those who underwent surgery were aged ≥65, in contrast to 34.1% of non-utilizers (p < 0.001).

Surgery utilization varied significantly by neighborhood socioeconomic status (DCI). A higher proportion of utilizers resided in prosperous (24.8%) and comfortable (28.5%) neighborhoods compared to non-utilizers (21.2% and 26.8%, respectively), while fewer came from distressed neighborhoods (12.5%) compared to non-utilizers (16.5%) (p < 0.001).

Racial and ethnic differences were also observed. Black individuals accounted for 46.9% of surgery recipients versus 39.6% of non-utilizers, while White individuals made up 46.1% of utilizers compared to 54.8% of non-utilizers (p < 0.001).

Insurance type differed substantially between groups. MBS patients were far more likely to have private insurance (70.7%) compared to non-utilizers (31.3%), and less likely to have Medicare (9.6%) compared to non-utilizers (45.1%) (p < 0.001).

Women were significantly more likely to undergo surgery, comprising 82.6% of the surgery group compared to 63.3% of non-utilizers (p < 0.001).

Surgery recipients were more likely to have Class II obesity (77.8%), while non-utilizers were more likely to fall into Class I (41.3%) (p < 0.001).

Fewer MBS recipients had severe comorbidities (Charlson Comorbidity Index >2), reported in 18.8% compared to 41.4% of non-utilizers (p < 0.001).

There were no clinically significant differences in urban-rural residence across groups, although differences reached statistical significance (Table 1).

### Baseline characteristics of study participants undergoing metabolic and bariatric surgery (MBS), stratified by DCI quintiles

Table 2 presents baseline characteristics of the 169 026 individuals who underwent MBS, stratified by DCI quintiles. Compared with the most prosperous quintile, patients in distressed neighborhoods were younger (27.0% aged 18–44 years vs. 20.9% in prosperous areas) and less likely to be aged ≥65 years (28.1% vs. 38.8%). Insurance coverage also diverged: Medicaid covered nearly one-third of patients from distressed communities (31.7%), whereas only 11.9% of patients in prosperous areas had Medicaid, and private insurance was far more common in prosperous communities (41.1% versus 20.6%). These age and insurance gradients illustrate that socioeconomically disadvantaged neighborhoods represent a younger, publicly insured case mix, which may partially explain observed disparities in surgical utilization. Distressed quintiles also had higher proportions of Black patients (62.1% vs. 24.7%), greater comorbidity burden (65.5% with CCI > 2 vs. 61.4%), and more severe obesity (64.6% vs. 56.4% with obesity class II), while prosperity was associated with a suburban/rural residence pattern that contrasted sharply with the predominantly urban distribution (68.1%) in distressed areas. These findings underscore the need to account for demographic and insurance differences when interpreting associations between neighborhood distress and metabolic-bariatric surgery outcomes (Table 2).

### Factors associated with MBS utilization in Maryland

Multivariable logistic regression revealed significant associations between neighborhood socioeconomic status, demographic characteristics, insurance type, comorbidities, and MBS in Maryland between 2018 and 2020 (Table 3).

**Table 1. Baseline Characteristics of Metabolic and Bariatric Surgery Eligible Individuals by Utilization Status.**

| Bariatric Surgery Utilization | Total Population | Bariatric surgery Utilizers | Non-Utilizers | Chi-Square | P |
|---|---|---|---|---|---|
| | (N = 169,026) | (N = 11,963) | (n = 157,063) | | |
| Age (Yr.) | 55.8 ± 15.4 | 44.4 ± 11.8 | 56.7 ± 15.3 | t = 85.83 | <0.001 |
| **Age (Yr.)** | | | | 9.80E+03 | <0.001 |
| 18-44Yr. | 77,016 (25.9%) | 9,929 (51.2%) | 67,087 (24.2%) | | |
| 445−64Yr. | 124,219 (41.8%) | 8,491 (43.7%) | 115,728 (41.7%) | | |
| ≥65Yr. | 95,666 (32.2%) | 992 (5.1%) | 94,674 (34.1%) | | |
| **Distressed Communities Index** | | | | 189.1007 | <0.001 |
| Prosperous | 33793 (21.5%) | 2757 (24.8%) | 31036 (21.2%) | | |
| Comfortable | 42301 (26.9%) | 3168 (28.5%) | 39133 (26.8%) | | |
| Mid-Tier | 34,509 (22.0%) | 2,264 (20.4%) | 32245 (22.1%) | | |
| At-Risk | 21,090 (13.4%) | 1,526 (13.7%) | 19,564 (13.4%) | | |
| Distressed | 25,499 (16.2%) | 1,389 (12.5%) | 24,110 (16.5%) | | |
| **Race/Ethnicity** | | | | 339.7864 | <0.001 |
| White | 90673 (54.2%) | 5398 (46.1%) | 85275 (54.8%) | | |
| Black | 67145 (40.1%) | 5502 (46.9%) | 61643 (39.6%) | | |
| Hispanic | 6259 (3.7%) | 557 (4.8%) | 5702 (3.7%) | | |
| **Other** | 3361 (2.0%) | 266 (2.2%) | 3095 (1.9%) | | |
| **Insurance** | | | | 8.50E+03 | |
| Self-Pay | 1976 (1.2%) | 36 (0.3%) | 1940 (1.2%) | | |
| Medicare | 71987 (42.6%) | 1148 (9.6%) | 70839 (45.1%) | | |
| Medicaid | 33001 (19.5%) | 2165 (18.1%) | 30836 (19.6%) | | <0.001 |
| Private | 57656 (34.1%) | 8451 (70.7%) | 49205 (31.3%) | | |
| Other | 4346 (2.6%) | 273 (1.5%) | 4092 (2.6%) | | |
| **Charlson Comorbidity Index** | | | | 1.00E+03 | |
| CCI 0 | 7216 (14.3%) | 986 (29.7%) | 6230 (13.2%) | | |
| Index (1,2) | 23126 (45.8%) | 1713 (51.6%) | 21413 (45.4%) | | |
| Index >2 | 20166 (39.9%) | 623 (18.8%) | 19543 (41.4%) | | <0.001 |
| **Sex** | | | | 1.80E+03 | |
| Male | 59761 (35.4%) | 2087 (17.5%) | 57674 (36.7%) | | |
| Female | 109254 (64.6%) | 9876 (82.6%) | 99378 (63.3%) | | |
| **Obesity Class** | | | | 1.70E+03 | <0.001 |
| Class I | 67472 (39.9%) | 2660 (22.2%) | 64812 (41.3%) | | |
| Class II | 101554 (60.1%) | 9303 (77.8%) | 92251 (58.7%) | | |
| **Urban** | | | | 49.968 | < 0.001 |
| Rural | 13,122 (8.2%) | 1,023 (9.2%) | 12,099 (8.2%) | | |
| Small Town | 10,833 (6.8%) | 780 (7.0%) | 10,053 (6.8%) | | |
| Suburban | 105,905 (66.4%) | 7,490 (67.4%) | 98,415 (66.3%) | | |
| Urban | 29,700 (18.6%) | 1,816 (16.4%) | 27,884 (18.8%) | | |

Baseline characteristics of study participants undergoing metabolic and bariatric surgery (MBS), stratified by Distressed Communities Index (DCI) quintiles. Values are counted with column percentages in parentheses. P-values were obtained from χ² or ANOVA tests as appropriate.

Compared to individuals from prosperous neighborhoods, the odds of undergoing MBS were significantly lower for those in mid-tier (OR = 0.743, 95% CI: 0.698–0.792) and distressed neighborhoods (OR = 0.697, 95% CI: 0.639–0.760). A similar trend was seen for individuals in comfortable (OR = 0.893, 95% CI: 0.842–0.946) and at-risk neighborhoods (OR = 0.893, 95% CI: 0.832–0.959).

**Table 2. Baseline Characteristics of Metabolic and Bariatric Surgery Patients Stratified by Distressed Communities Index Quintiles.**

| (N = 169,026) | Prosperous (n = 34,494) | Comfortable (n = 42,999) | Mid-Tier (n = 35,022) | At-Risk (n = 21,320) | Distressed (n = 25,725) | p-value |
|---|---|---|---|---|---|---|
| Age, year, Mean ± SD | 58.1 ± 15.8 | 57.0 ± 15.7 | 55.5 ± 16.0 | 55.3 ± 15.8 | 54.4 ± 15.5 | |
| **Age** | | | | | | <0.01 |
| 18-44yrs. | 7,204 (20.9%) | 9,606 (22.3%) | 9,044 (25.8%) | 5,506 (25.8%) | 6,947 (27.0%) | |
| 45-64yrs. | 13,892 (40.3%) | 18,033 (41.9%) | 14,847 (42.4%) | 9,113 (42.7%) | 11,539 (44.9%) | |
| >64 yrs. | 13,398 (38.8%) | 15,360 (35.7%) | 11,131 (31.8%) | 6,701 (31.4%) | 7,239 (28.1%) | |
| **Sex** | | | | | | <0.01 |
| Male | 13,191 (38.2%) | 15,345 (35.7%) | 11,807 (33.7%) | 7,198 (33.8%) | 7,929 (30.8%) | |
| Female | 21,301 (61.8%) | 27,651 (64.3%) | 23,214 (66.3%) | 14,121 (66.2%) | 17,793 (69.2%) | |
| **Race/Ethnicity** | | | | | | <0.01 |
| White | 23,644 (69.5%) | 25,364 (59.6%) | 15,401 (44.4%) | 11,060 (52.2%) | 9,035 (35.3%) | |
| Black | 68,417 (24.7%) | 14,791 (34.8%) | 16,440 (47.4%) | 8,890 (41.9%) | 15,914 (62.1%) | |
| Hispanic | 1,102 (3.2%) | 1,462 (3.4%) | 2,186 (6.3%) | 883 (4.2%) | 365 (1.4%) | |
| Other | 857 (2.5%) | 923 (2.2%) | 689 (2.0%) | 364 (1.72%) | 311 (1.2%) | |
| **Insurance** | | | | | | <0.01 |
| Self-Pay | 270 (0.8%) | 433 (1.0%) | 547 (1.6%) | 267 (1.2%) | 211 (0.8%) | |
| Medicare | 15,156 (44.0%) | 18,846 (43.9%) | 14,479 (41.4%) | 9,354 (43.9%) | 11,658 (45.4%) | |
| Medicaid | 4,096 (11.9%) | 6,961 (16.2%) | 7,766 (22.2%) | 5,144 (24.1%) | 8,138 (31.7%) | |
| Private | 14,189 (41.2%) | 15,834 (36.9%) | 11,509 (32.9%) | 6,194 (29.1%) | 5,300 (20.6%) | |
| Other | 752 (2.2%) | 864 (2.0%) | 696 (2.0%) | 345 (1.6%) | 384 (1.5%) | |
| **Urban Classification** | | | | | | <0.01 |
| Rural | 4,196 (12.2%) | 3,675 (8.5%) | 2,838 (8.1%) | 2,036 (9.5%) | 377 (1.5%) | |
| Small Town | 1,468 (4.3%) | 3,195 (7.4%) | 1,685 (4.8%) | 462 (2.2%) | 4,023 (15.6%) | |
| Suburban | 26,850 (77.8%) | 34,501 (80.2%) | 26,332 (75.2%) | 14,421 (67.6%) | 3,801 (14.8%) | |
| Urban | 1,980 (5.7%) | 1,628 (3.8%) | 4,167 (11.9%) | 4,401 (20.6%) | 17,524 (68.1%) | |
| **Bariatric Surgery** | | | | | | <0.01 |
| No Surgery | 37,735 (92.0%) | 39,830 (92.6%) | 32,756 (93.5%) | 19,794 (92.8%) | 24,336 (94.6%) | |
| Surgery | 2,759 (8.0%) | 3,169 (7.4%) | 2,266 (6.5%) | 1,526 (7.2%) | 1,389 (5.4%) | |
| **CCI** | | | | | | <0.01 |
| CCI 0 | 5,682 (16.5%) | 7,076 (16.5%) | 6,193 (17.7%) | 3,469 (16.3%) | 3,641 (14.1%) | |
| Index (1,2) | 7,631 (22.1%) | 9,246 (21.5%) | 7,422 (21.2%) | 4,415 (20.7%) | 5,245 (20.4%) | |
| Index >2 | 21, 181 (61.4%) | 26,677 (62.0%) | 21,407 (61.1%) | 13,436 (63.0%) | 16,839 (65.5%) | |
| **Obesity Class** | | | | | | <0.01 |
| Class I | 15,053 (43.6%) | 17,677 (41.1%) | 13,998 (40.0%) | 8,156 (38.3%) | 9,117 (35.4%) | |
| Class II | 19,441 (56.4%) | 25,322 (58.9%) | 21,024 (60.0%) | 13,164 (61.7%) | 16, 608 (64.6%) | |

Counts (N) and column percentages (%) summarize demographic variables, health insurance types, urban classification, Charlson comorbidity index and obesity class across quintiles of the Distressed Communities Index (DCI). Distressed neighborhoods contain a younger age distribution and a higher prevalence of Medicaid or uninsured patients compared with prosperous areas, where patients tend to be older and have private insurance. These differences in age and insurance coverage may act as potential confounders when examining associations between neighborhood socioeconomic status and metabolic–bariatric surgery utilization. P-values were calculated using $\chi^2$ or ANOVA tests for comparisons across DCI categories.

Age was inversely associated with surgery utilization. Patients aged 45–64 years had significantly lower odds of receiving MBS compared to those aged 18–44 (OR = 0.463, 95% CI: 0.441–0.486), and those aged ≥65 years had dramatically lower odds (OR = 0.142, 95% CI: 0.128–0.158). Females were more than twice as likely as males to undergo MBS (OR = 2.620, 95% CI: 2.482–2.765).

**Table 3. Factors associated with MBS Utilization in Maryland (2018-2020).**

| Bariatric Surgery Utilization | Odds Ratio | Std. Err. | z | P>z | 95% CI | |
|---|---|---|---|---|---|---|
| **Distressed Communities Index** | | | | | | |
| Prosperous | Reference | | | | | |
| Comfortable | 0.893 | 0.026 | −3.840 | <0.001 | 0.842 | 0.946 |
| Mid-Tier | 0.743 | 0.024 | −9.180 | <0.001 | 0.698 | 0.792 |
| At-Risk | 0.893 | 0.033 | −3.100 | 0.002 | 0.832 | 0.959 |
| Distressed | 0.697 | 0.031 | −8.160 | <0.001 | 0.639 | 0.760 |
| **Age (Yr.)** | | | | | | |
| 18-44Yr. | Reference | | | | | |
| 45/64Yr. | 0.463 | 0.012 | −30.820 | <0.001 | 0.441 | 0.486 |
| >64Yr. | 0.142 | 0.008 | −35.300 | <0.001 | 0.128 | 0.158 |
| **Female (ref. Male)** | 2.620 | 0.072 | 34.890 | <0.001 | 2.482 | 2.765 |
| **Race and Ethnicity** | | | | | | |
| White | Reference | | | | | |
| Black | 1.297 | 0.031 | 10.930 | <0.001 | 1.238 | 1.359 |
| Hispanic | 1.114 | 0.059 | 2.060 | 0.040 | 1.005 | 1.235 |
| Other | 1.003 | 0.073 | 0.040 | 0.971 | 0.869 | 1.158 |
| **Insurance** | | | | | | |
| Self-Pay | Reference | | | | | |
| Medicare | 1.726 | 0.313 | 3.010 | 0.003 | 1.210 | 2.463 |
| Medicaid | 2.783 | 0.497 | 5.730 | <0.001 | 1.961 | 3.951 |
| Private | 8.611 | 1.531 | 12.110 | <0.001 | 6.077 | 12.200 |
| Other | 3.160 | 0.627 | 5.800 | <0.001 | 2.142 | 4.661 |
| **Charlson Comorbidity Index** | | | | | | |
| CCI 0 | Reference | | | | | |
| Index (1,2) | 1.528 | 0.044 | 14.790 | <0.001 | 1.444 | 1.616 |
| Index >2 | 1.013 | 0.031 | 0.430 | 0.664 | 0.955 | 1.075 |
| **Obesity Class III (ref.class II)** | 2.252 | 0.056 | 32.610 | 0.000 | 2.145 | 2.365 |
| **Urbanicity** | | | | | | |
| Rural | Reference | | | | | |
| Small Town | 1.003 | 0.055 | 0.060 | 0.951 | 0.901 | 1.118 |
| Suburban | 0.741 | 0.029 | −7.730 | <0.001 | 0.687 | 0.800 |
| Urban | 0.687 | 0.035 | −7.460 | <0.001 | 0.622 | 0.758 |
| _cons | 0.011 | 0.002 | −24.380 | <0.001 | 0.008 | 0.016 |

In terms of race and ethnicity, Black individuals had significantly higher odds of surgery compared to Whites (OR = 1.297, 95% CI: 1.238–1.359), as did Hispanic individuals (OR = 1.114, 95% CI: 1.005–1.235). No significant differences were observed among individuals categorized as "Other."

Insurance type was strongly associated with utilization. Compared to self-pay patients, those with private insurance had the highest odds of surgery (OR = 8.611, 95% CI: 6.077–12.200), followed by those with other insurance (OR = 3.160), Medicaid (OR = 2.783), and Medicare (OR = 1.726) (all p < 0.01).

Patients with Obesity Class III were more than twice as likely to receive surgery compared to those with Class II obesity (OR = 2.252, 95% CI: 2.145–2.365).

Comorbidity burden was also associated: patients with a CCI of 1–2 had increased odds (OR = 1.528, 95% CI: 1.444–1.616), while CCI > 2 was not significantly associated with utilization.

Finally, urbanicity revealed geographic disparities. Compared to patients in rural areas, those in urban (OR = 0.687) and suburban (OR = 0.741) areas were significantly less likely to undergo surgery, while no significant difference was seen for residents of small towns (Table 3).

**Sensitivity analysis using the area deprivation index**

As a sensitivity analysis, the ADI was used in place of the DCI to assess neighborhood socioeconomic status (Table 4). Individuals from moderately deprived neighborhoods (25th–75th percentile) had 18% lower odds of undergoing MBS compared to those from the least deprived areas (OR = 0.821, 95% CI: 0.778–0.865, p<0.001), while those from the most deprived neighborhoods (>75th percentile) had 21% lower odds (OR = 0.790, 95% CI: 0.747–0.835, p<0.001). These findings confirm a consistent socioeconomic gradient in MBS utilization, reinforcing the robustness of the primary results (Table 4).

**Race as a moderator of the association between neighborhood distress and MBS utilization**

The association between neighborhood socioeconomic status—measured using the DCI and MBS utilization showed variation by race. Across all DCI quintiles, Black patients were more likely than White patients to undergo bariatric surgery. The marginal effect of being Black versus White increased with community distress, from 0.2% (95% CI: –0.4%, 0.9%) in the most prosperous neighborhoods to 1.8% (95% CI: 1.2%, 2.3%) in the most distressed communities. In the middle three quintiles (Comfortable, Mid-Tier, and At-Risk), the marginal effects ranged from 1.0% to 1.3%, all statistically significant except in the most prosperous areas. (Fig 1). Fig 1 illustrates the predicted marginal effects of community distress on MBS utilization by race, highlighting that the difference in predicted MBS utilization between Black and White patients widens as neighborhood distress increases.

## Discussion

This study yielded three important findings. First, despite meeting eligibility criteria, only 7.0% of individuals underwent MBS though higher than the national average of 1% but still markedly low, highlighting a substantial gap between clinical eligibility and treatment uptake. Second, there was a clear socioeconomic gradient in MBS utilization. Individuals residing in socioeconomically distressed neighborhoods, as measured by the DCI, were significantly less likely to undergo surgery compared to those in more affluent areas. Surgery rates were highest among individuals in prosperous communities and declined progressively across increasing levels of neighborhood disadvantage. Third, race and ethnicity moderate the association between neighborhood socioeconomic status and MBS utilization. Black individuals had significantly higher utilization than White individuals across all DCI quintiles, with the largest difference observed in the most distressed neighborhoods. For other racial and ethnic groups, race did not significantly modify the relationship between neighborhood distress and surgery utilization.

These findings highlight a complex interaction between race, place, and access to MBS. The observed decline in utilization among residents of more distressed communities aligns with previous literature indicating that lower neighborhood

**Table 4. Association between Area Deprivation Index and Metabolic and Bariatric Surgery Utilization in Maryland (2018–2020).**

| Bariatric Surgery Utilization | Odds Ratio | Std. Err. | z | P>z | 95% CI | |
|---|---|---|---|---|---|---|
| **Area Deprivation Index** | | | | | | |
| < 25th Percentile | Reference | | | | | |
| 25-75th Percentile | 0.821 | 0.024 | −4.880 | <0.001 | 0.778 | 0.865 |
| > 75th Percentile | 0.790 | 0.028 | −1.490 | <0.001 | 0.747 | 0.835 |

Model adjusted for age, sex, race/ethnicity, insurance type, pre-existing comorbidity, obesity classification and urban classification.

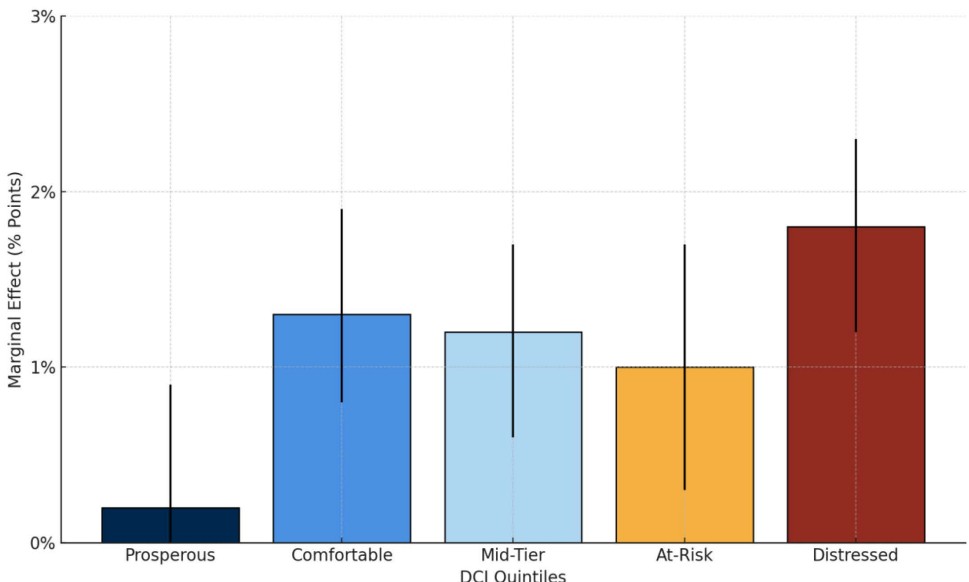

**Fig 1. Marginal effects of race (Black vs. White) on bariatric surgery utilization by neighborhood distress (DCI), Maryland 2018–2020.**

socioeconomic status is associated with reduced access to elective surgical care [30–33]. Contributing factors may include limited healthcare infrastructure, fewer specialty providers, transportation challenges, and lower health literacy in underserved communities [30,34,35]. Despite Maryland's All-Payer Model which minimizes variation in hospital reimbursement rates and aims to reduce healthcare disparities these structural barriers persist and continue to shape access to surgical obesity care [36,37].

The higher odds of MBS among Black individuals across all levels of neighborhood distress contrasts with national patterns that typically show lower utilization among Black patients [13,38,39]. This divergence may reflect a combination of increased clinical need, greater provider awareness of obesity disparities, and public health or hospital-level efforts to promote equitable access [40]. In addition, obesity is more prevalent and often more severe among Black individuals, potentially leading to higher referral rates [41,42]. The consistent pattern across all DCI quintiles, with a notable intensification in distressed areas, indicates that targeted equity-promoting strategies may be working more effectively in these settings. In contrast, Hispanic patients only experienced greater utilization in the most distressed neighborhoods, indicating persistent disparities in more socioeconomically advantaged areas [43]. Cultural and linguistic barriers, immigration-related concerns, and differences in healthcare-seeking behavior may contribute to these patterns and warrant targeted intervention [44–47].

Additional factors may contribute to disparities beyond those captured in the present study. Referral patterns are influenced by provider-level factors, including obesity training, implicit and explicit racial or weight bias, and perceptions of surgical risk [48–50]. Previous studies highlight that primary care providers' implicit or explicit racial and weight bias, their knowledge of MBS indications, and current reimbursement models may shape the likelihood of referring patients of different race/ethnicity, insurance coverage and socioeconomic status for multidisciplinary evaluation [51–53]. Provider bias can also affect surgeons' decisions once patients reach specialty care [54]. Other studies have documented differences in perioperative management attributable to provider racial bias, such as higher rates of prophylactic interventions among Black patients despite lower risk factors [55,56]. The geographic availability of multidisciplinary weight centers and surgeons capable of offering MBS, as well as transportation barriers, further limits access [57,58]. Systematic reviews and qualitative studies report that long distances to bariatric centers, unreliable public transportation, and the cost and time associated with travel deter referrals and follow-up, particularly for patients from rural or distressed neighborhoods [59].

Insurance status may function as a mediator rather than a mere confounder in the relationship between neighborhood disadvantages and MBS access. Medicaid insurance is often used as a proxy for socioeconomic deprivation, and studies have found that Medicaid patients experience lower weight loss, longer hospital stays and higher complication rates after MBS compared with privately insured patients [50]. Differences in insurance coverage can influence whether providers refer patients, whether insurers approve the procedure, and the level of resources available for postoperative support [40]. Exploring these mediation pathways could illuminate how socioeconomic disadvantage translates into disparities in MBS utilization.

Although our models adjusted for age and insurance, our post-hoc analysis confirmed that individuals' age and insurance type differed across DCI categories. The distressed neighborhoods have a younger age structure and a higher proportion of Medicaid coverage, whereas prosperous communities have more older residents and higher private insurance coverage. Such compositional differences could mediate or confound the observed association between neighborhood disadvantage and MBS utilization. To fully disentangle the roles of age and insurance in the relationship between socioeconomic context and bariatric surgery uptake, future work should explicitly model these mediating pathways. Efforts to improve equity should also consider how age, insurance and neighborhood socioeconomic status interact to shape access to care.

Together, these results underscore the importance of intersectional approaches to understanding disparities in MBS. While race and neighborhood disadvantage independently affect access to care, their interaction reveals deeper structural inequities that influence treatment uptake [60–62]. Tailored strategies are needed to expand access in high-poverty neighborhoods and to ensure that underrepresented racial and ethnic groups are not overlooked in referral and treatment pathways [63].

## Strengths and limitations

This population-based study has several strengths. The use of the DCI, a multidimensional and validated measure of neighborhood socioeconomic status, allowed us to operationalize community-level deprivation across multiple domains. The Maryland All-Payer Model provided a policy context with minimal variation in hospital reimbursement, enabling us to focus on non-financial determinants of MBS utilization. In addition, the use of a large and racially/ethnically diverse administrative dataset enhances the precision of estimates and the internal validity of the findings.

However, the study has important limitations that affect how the results should be interpreted. First, the cross-sectional design precludes determination of temporal ordering and causality; exposures and outcomes were measured at the same time, making it impossible to know whether neighborhood socioeconomic status influences MBS utilization or whether other unmeasured factors drive both. Cross-sectional surveys are prone to reverse causation, so associations must be interpreted as descriptive rather than causal. Second, although administrative inpatient data allowed for population-level coverage, they lacked information on important individual-level factors such as health beliefs, cultural perceptions, obesity severity, comorbidities, patient preferences and provider referrals that may influence both the likelihood of being offered surgery and the decision to undergo it. In addition, the dataset did not capture outpatient evaluations, insurance precertification processes or procedures performed in out-of-state hospitals, which could bias our estimates of access. Third, the DCI is assigned at the zip-code level; this spatial aggregation may mask substantial within-area heterogeneity, and residual confounding by other community factors (e.g., rurality, healthcare infrastructure) likely remains. Fourth, because the study population was drawn from a single state operating under a unique reimbursement model, the results may not generalise to states with different healthcare systems or insurance policies. Finally, despite adjusting for multiple covariates, unmeasured confounders such as provider weight or racial bias, referral practices, and patient comorbidity profiles could mediate or confound the observed associations. Together, these limitations mean that our findings should not be interpreted as evidence of a causal effect of neighborhood socioeconomic disadvantage on MBS utilization; rather, they highlight associations that warrant further investigation in longitudinal and mixed-methods studies.

## Conclusion

This study provides evidence of an association between neighborhood socioeconomic status and metabolic and MBS utilization in Maryland. Individuals living in the most distressed communities were less likely to undergo surgery compared to those in more affluent neighborhoods. Importantly, patterns varied by race and ethnicity: Black individuals demonstrated consistently higher utilization rates than White individuals across all levels of community distress, with the largest difference observed in the most disadvantaged neighborhoods. These findings underscore the complex interplay of socioeconomic and racial/ethnic factors in MBS utilization and highlight the need for further research to understand underlying mechanisms and inform future strategies aimed at addressing disparities.

## Author contributions

**Conceptualization:** Oluwasegun Akinyemi, Kakra Hughes, Jie Chen, Kellee White-Whilby.

**Data curation:** Oluwasegun Akinyemi, Mojisola Fasokun, Kakra Hughes, Dahai Yue, Craig Scott Fryer, Jie Chen, Kellee White-Whilby.

**Formal analysis:** Oluwasegun Akinyemi, Dahai Yue, Jie Chen, Kellee White-Whilby.

**Funding acquisition:** Oluwasegun Akinyemi, Terrence Fullum, Kakra Hughes.

**Investigation:** Oluwasegun Akinyemi, Terrence Fullum, Mojisola Fasokun, Kakra Hughes, Dahai Yue, Craig Scott Fryer, Kellee White-Whilby.

**Methodology:** Oluwasegun Akinyemi, Craig Scott Fryer, Jie Chen, Kellee White-Whilby.

**Project administration:** Oluwasegun Akinyemi, Terrence Fullum, Mojisola Fasokun, Kakra Hughes, Dahai Yue, Craig Scott Fryer, Jie Chen, Kellee White-Whilby.

**Resources:** Oluwasegun Akinyemi, Terrence Fullum, Mojisola Fasokun, Kakra Hughes, Dahai Yue, Craig Scott Fryer, Jie Chen, Kellee White-Whilby.

**Software:** Oluwasegun Akinyemi, Terrence Fullum, Mojisola Fasokun, Kakra Hughes, Dahai Yue, Craig Scott Fryer, Jie Chen, Kellee White-Whilby.

**Supervision:** Oluwasegun Akinyemi, Terrence Fullum, Kakra Hughes, Dahai Yue, Craig Scott Fryer, Jie Chen, Kellee White-Whilby.

**Validation:** Oluwasegun Akinyemi, Terrence Fullum, Kakra Hughes, Dahai Yue, Craig Scott Fryer, Jie Chen, Kellee White-Whilby.

**Visualization:** Oluwasegun Akinyemi, Jie Chen, Kellee White-Whilby.

**Writing – original draft:** Oluwasegun Akinyemi, Mojisola Fasokun, Jie Chen, Kellee White-Whilby.

**Writing – review & editing:** Oluwasegun Akinyemi, Terrence Fullum, Mojisola Fasokun, Kakra Hughes, Dahai Yue, Craig Scott Fryer, Jie Chen, Kellee White-Whilby.

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
