## [Decision Letter · Decision Letter 0]

19 Sep 2025

Dear Dr. Akinyemi,

Thank you for submitting your manuscript to PLOS ONE. After careful consideration, we feel that it has merit but does not fully meet PLOS ONE’s publication criteria as it currently stands. Therefore, we invite you to submit a revised version of the manuscript that addresses the points raised during the review process.

https://journals.plos.org/plosone/s/submission-guidelines#loc-laboratory-protocols . Additionally, PLOS ONE offers an option for publishing peer-reviewed Lab Protocol articles, which describe protocols hosted on protocols.io. Read more information on sharing protocols at https://plos.org/protocols?utm_medium=editorial-email&utm_source=authorletters&utm_campaign=protocols .

We look forward to receiving your revised manuscript.

Kind regards,

Athanasios G. Pantelis

Academic Editor

PLOS ONE

Journal Requirements:

2. Please note that your Data Availability Statement is currently missing [the repository name and/or the DOI/accession number of each dataset OR a direct link to access each database]. If your manuscript is accepted for publication, you will be asked to provide these details on a very short timeline. We therefore suggest that you provide this information now, though we will not hold up the peer review process if you are unable.

Additional Editor Comments:

Please address all reviewers' comments before resubmitting your manuscript fro publication to PLOS ONE.

Reviewers' comments:

Reviewer's Responses to Questions

**Comments to the Author**

1. Is the manuscript technically sound, and do the data support the conclusions?

Reviewer #1: Yes

Reviewer #2: Yes

2. Has the statistical analysis been performed appropriately and rigorously?

Reviewer #1: Yes

Reviewer #2: Yes

3. Have the authors made all data underlying the findings in their manuscript fully available?

Reviewer #1: Yes

Reviewer #2: Yes

4. Is the manuscript presented in an intelligible fashion and written in standard English?

Reviewer #1: Yes

Reviewer #2: Yes

Reviewer #1: Dear Authors,

I read your manuscript carefully and found it timely, policy-relevant, that focused on a health equity issue. However, several areas could be strengthened in terms of clarity, framing, methodological detail, and interpretation of race-specific findings.

1- Please use "Metabolic and Bariatric surgery (MBS)" instead of bariatric surgery, alone.

2- I suggest expanding the discussion about other important factors such as referral bias, racism, hospital distance from referral points.

3- Discuss more how insurance can have effect on nSES. Could insurance status be a mediator rather than a confounder factor?

4- Please add a brief description of Figure 1 in the results or discussion to make it easier to understand.

Reviewer #2: I commend the authors for a very thorough work and interesting findings that could help healthcare directed to certain populations regarding Bariatric Surgery in particular. My only points regarding the results and possible confounding factors are age and insurance coverage across neighborhood socioeconomic status which are not analysed per DCI category. This of course could influence the results either way since in more distressed neighborhoods there is less likelihood of private insurance but probably younger population so it would be complex to say which way the influence would be more profound.

**Do you want your identity to be public for this peer review?** For information about this choice, including consent withdrawal, please see our Privacy Policy

Reviewer #1: **Yes: ** Mohammad Kermansaravi

Reviewer #2: No

---

## [Author Response · Author response to Decision Letter 1]

24 Oct 2025

Dear Editor and Reviewers,

Thank you for your thoughtful evaluation of our manuscript and for the opportunity to revise it. We appreciate the insightful comments from the Academic Editor and both reviewers, and we have carefully addressed each point. Below, we provide a point‑by‑point response and a summary of changes made to the revised manuscript.

Editor’s Comments

1. Use of IFSO‑compliant nomenclature. We have replaced the term “bariatric surgery” with “metabolic and bariatric surgery (MBS)” throughout the manuscript, including in the title, abstract, tables, and narrative text. The abbreviation MBS is now defined at first use and used consistently thereafter.

2. STROBE checklist. A completed STROBE checklist is provided as a separate file. In it, we indicate the page and section in the manuscript where each reporting item is addressed.

3. Causal language. We revised the manuscript to avoid causal or policy‑prescriptive wording, reflecting the observational and cross‑sectional nature of our study. Statements that previously implied causality (for example, “effect of nSES on MBS use”) now describe associations (e.g., “association between nSES and MBS utilization”), and we emphasized that causal inferences cannot be drawn from these data.

4. Limitations. The limitations section has been expanded and now more clearly articulates how unmeasured confounding, selection bias, measurement limitations, and the cross‑sectional design may influence interpretation of our findings. We explicitly note that age and insurance distributions vary across socioeconomic strata and may confound the observed associations, and that our reliance on administrative data may lead to misclassification of comorbidities and procedures (Page 16-17, Lines 332-360).

In addition, we have updated the Data Availability Statement and checked the reference list for completeness and currency, replacing any outdated citations.

Reviewer #1

1. Use “Metabolic and Bariatric Surgery (MBS)” instead of “bariatric surgery”. As noted above, the nomenclature has been updated throughout the manuscript, including the title and table captions.

2. Expand discussion of referral bias, racism, and hospital distance. The Discussion now includes a paragraph addressing potential referral bias and structural racism in access to surgical care, citing relevant literature. We also discuss how geographical distance to MBS‑capable hospitals may reduce utilization among individuals in distressed neighborhoods (Discussion, Page 15, Lines 293-306).

3. Insurance as mediator or confounder. We added text to the Discussion noting that insurance coverage may function as both a mediator and a confounder in the relationship between neighborhood socioeconomic status and MBS utilization. We caution that Medicaid and uninsured status can both reflect socioeconomic disadvantage and limit access to surgery, and we note that further causal mediation analyses are beyond the scope of our cross‑sectional study (Page 15, Lines 307-323).

4. Description of Figure 1. The Results section now briefly describes Figure 1, highlighting the geographic distribution of neighborhood distress across Maryland and the corresponding variation in MBS utilization (Page 13, Lines 257-259).

Reviewer #2

We appreciate the reviewer’s observation that age and insurance coverage may confound our results. To address this, we conducted additional analyses of age, insurance and other study variables distributions across DCI quintiles and present these results in a new Table 2 (Pages 10-11, Lines 193-209, Page 26-27). The table shows that distressed neighborhoods have a higher proportion of younger adults (27.0 % aged 18–44 years vs. 20.9 % in prosperous areas) and a higher prevalence of Medicaid coverage (31.6 % vs. 11.9 %), whereas private insurance is more common in prosperous communities (41.1 % vs. 20.6 %). These differences are summarized in the Results (Page 10, Lines 193-209) and discussed in the context of potential confounding in the Discussion (Page 15, Lines 307-324).

We thank the reviewers again for their constructive feedback. We believe that these revisions have strengthened the manuscript and clarified the interpretation of our findings. We hope the revised submission meets the standards of PLOS ONE and look forward to your decision.

Sincerely,

Oluwasegun Akinyemi, MD, MSc., PhD (on behalf of all authors)

---

## [Editor Report · Decision Letter 1]

28 Oct 2025

Variation of Metabolic and Bariatric Surgery Utilization by Neighborhood Socioeconomic Status in Maryland

PONE-D-25-41261R1

Dear Dr. Akinyemi,

We’re pleased to inform you that your manuscript has been judged scientifically suitable for publication and will be formally accepted for publication once it meets all outstanding technical requirements.

Kind regards,

Athanasios G. Pantelis

Academic Editor

PLOS ONE
---

## [Editor Report · Acceptance letter]

PONE-D-25-41261R1

PLOS ONE

Dear Dr. Akinyemi,

I'm pleased to inform you that your manuscript has been deemed suitable for publication in PLOS ONE. Congratulations! Your manuscript is now being handed over to our production team.

Kind regards,

on behalf of

Dr. Athanasios G. Pantelis

Academic Editor

PLOS ONE